# Sonic irrigation for removal of calcium hydroxide in the apical root canal: A micro-CT and light-coupled tracking analysis

**Wonjoon Moon[1‡], Shin Hye Chung**[1‡]**, Juhea Chang**[2]*****

**1** Department of Dental Biomaterials Science, School of Dentistry and Dental Research Institute, Seoul National University, Jongno-gu, Seoul, Republic of Korea, **2** National Dental Care Center for Persons with Special Needs, Seoul National University Dental Hospital, Jongno-gu, Seoul, Republic of Korea

‡ WM and SHC are contributed equally to this work as the co-first authors.
* juhchang@snu.ac.kr

**Data Availability Statement:** All relevant data are within the manuscript and its Supporting Information files.

**Funding:** This work was supported by Creative-Pioneering Researchers Program through Seoul

## Abstract

### Objective

This study aimed to evaluate the efficacy of three sonic irrigation systems for removal of calcium hydroxide dressing from the apical root canal.

### Materials and methods

A total of 96 single-rooted teeth in three categories of root canal curvatures (straight: 0–5˚, moderate: 6–20˚, and severe: > 20˚) were allocated to four groups: conventional needle irrigation, EndoActivator, EQ-S, and Vibringe. The root canals were instrumented using Protaper NEXT and filled with calcium hydroxide. After removal of calcium hydroxide, the remaining volume of calcium hydroxide was measured by micro-CT analysis. Data were compared among root canal curvatures and irrigation systems using the Kruskal-Wallis test and Mann-Whitney test ($p < .05$). The oscillating range of each irrigation system was measured using light-coupled motion tracking.

### Results

The volumes of calcium hydroxide remaining in the canals with severe curvature were significantly higher than in those of straight curvature ($p < .05$). In the canals of moderate or severe curvature, EQ-S showed the highest removal percentage, followed by EndoActivator, Vibringe, and needle irrigation ($p < .05$). Light-coupled tracking showed the largest oscillating range in EQ-S ($p < .05$).

### Conclusions

Sonically activated irrigation systems with a flexible tip can be beneficial for calcium hydroxide intracanal dressing removal in the curved apical canals.

National University (SNU)(https://www.snu.ac.kr/) and the National Research Foundation of Korea (NRF) grant funded by the Korea government (MSIT) (No. 2020R1A2C1102316) The funders had no role in study design, data.

**Competing interests:** The authors have declared that no competing interests exist.

## Introduction

Calcium hydroxide is the most common intracanal medicament used during endodontic treatment [1]. It has diverse physical and biological advantages such as antibacterial effects, ability to dissolve tissue, promotion of hard tissue formation, reduction of bacterial toxic products, and healing of periapical tissues [2]. However, complete removal of calcium hydroxide might not clinically achievable, particularly in root canals with anatomical complexities [3]. Remnants of calcium hydroxide can hinder penetration of sealers into dentinal tubules and negatively affect sealing of canal filling materials [4].

For effective removal of calcium hydroxide, various technologies were incorporated as adjunctive methods to conventional needle irrigation by activating fluid movement within the canal space. Effective cleaning requires the ability to bring the irrigant in contact with the elements to be removed while not damaging the root dentin structures [5]. Ultrasonic devices were first introduced in mechanical debridement in endodontics and developed into passive ultrasonic irrigation (PUI) for enhancement of intracanal debridement [6]. PUI systems were designed to introduce a smooth file or wire with a noncutting tip into the canal, and energy transmission is controlled to avoid intentional contact with the canal walls. However, in a small constraint in the canal, wall contact is unavoidable, and free oscillation is inhibited, particularly in the apical roots with geometrical complexities [7]. Consequently, risks can arise such as needle binding within the irregular dentinal walls, deforming the root canal morphology, and weakening the apical constriction [8]. An alternative option, passive sonic activation, incorporates non-cutting plastic tips oscillating at much lower frequencies (150–200 Hz) compared to PUI (25,000–30,000 Hz). In addition, low electric current-assisted sonic irrigation has been introduced to even increase the efficiency compared to conventional sonic irrigation and PUI [9]. These sonic systems attempted to avoid excessive cavitation and produce sufficient fluid agitation that removes the smear layer and dislodges intraradicular biofilm [10].

Previous evaluations of activation systems for root canal irrigation have noted the major obstacle against efficient cleaning as complicated root canal anatomy [10, 11]. This limited cleaning effect is more pronounced below the root canal curvature narrowing to the apical constriction [12]. Considering the three-dimensional complexities of the apical third of the root canal system, clinical outcomes can be evaluated based on both qualitative and quantitative measurements [13]. Several studies have focused on apical third of root canals and have introduced micro-CT analysis as an outcome measurement regarding each step of clinical procedures: canal preparation and shaping [14, 15], removal of previously filled materials [16, 17], removal of hard tissue debris [3, 18], and final intracanal obturation [19, 20]. The removal of the calcium hydroxide dressing at the interappointment session is another critical part of ensuring clinical expertise and can be clearly confirmed by 3D image construction of the root canal structures.

In this study, we applied conventional needle irrigation and three types of sonic irrigation systems (EndoActivator, EQ-S, and Vibringe) for removal of intracanal calcium hydroxide and evaluated the volumes of calcium hydroxide remaining in the apical space among different root canal curvatures using micro-CT. We also measured oscillating ranges of sonically-activated irrigation tips by a light-coupled motion tracking. The null hypothesis of this study was that removal efficacy of calcium hydroxide at the apical root canal would not be affected by sonic irrigation systems.

## Materials and methods

### Preparation of specimens

Ninety-six extracted single-rooted teeth with completed apices and no visible caries, cracks, or other defects were used in the study. The Institutional Review Board of Seoul National

University School of Dentistry approved this study (IRB No. S-D20200031). All procedures performed in this study involving human teeth were in accordance with relevant guidelines and regulations of the institutional research committee, and informed consent was obtained from all individual participants. The teeth were decoronated and sectioned to a length of 12 mm. The roots of the teeth were ultrasonically cleaned and stored in 0.9 g/L thymol solution (Sigma-Aldrich, St. Louis, MO, USA) [21]. Micro-CT scans (Skyscan 1172, Bruker, Kontich, Belgium) were performed to obtain sagittal images of each root. The scanning parameters were 100kV and 100μA at the Al + Cu filter with an exposure time of 632ms. The pixel size was 30μm with a rotation of 0.70 and an average frame number of three. The 3D images were acquired by 3D reconstruction (NRecon, Bruker, Kontich, Belgium), modeling (CTAn, Bruker, Kontich, Belgium), and analysis (CTVol, Bruker, Kontich, Belgium). Based on the sagittal images of each root, the curvatures were calculated with the Schneider method [22]. Considering their root curvatures, the 96 teeth were assigned into one of three categories: straight (0–5˚), moderate (6–20˚), and severe (> 21˚) (n = 32/category). The 32 teeth in each category were then randomly distributed into four groups of irrigation systems: Group 1 (control), Group 2 (EndoActivator), Group 3 (EQ-S), and Group 4 (Vibringe) (n = 8/group). The sample size was determined based on a previous study that reported means and SDs for removal efficacy of calcium hydroxide using micro-CT imaging [23].

## Root canal preparation and calcium hydroxide filling

Canal patency was confirmed with a #K-10 file (K-file, Maillefer Instruments, Ballaigues, Switzerland) until the tip was just visible at the apical foramen [18]. The working lengths were determined as 1 mm less than that length. All canals were prepared with Ni-Ti files using a rotary system (X-Smart, Dentsply Maillefer, Ballaigues, Switzerland) up to the X3 file (Protaper Next, Dentsply Maillefer, Ballaigues, Switzerland), ISO size 30, and taper 0.07, resulting in standardized root canals. During instrumentation, the root canals were irrigated with 2 mL 1% NaOCl solution at each change of instrumentation. After canal preparation, the canal space was dried with paper points (Absorbent Paper Points, Meta Biomed, Cheongju, Korea). The tip of the syringe containing paste was inserted into the canal to 1 mm shorter than the binding point [23]. The 0.1 mL of calcium hydroxide paste (Calcipex II, Nippon Shika Yakuhin, Shimonoseki, Japan) was injected slowly with minimal pressure to fill the space. The calcium hydroxide paste used in this study consisted of calcium hydroxide, barium sulfate, and distilled water and is easy to handle as a root canal filling material [24]. The medicament was then condensed using a paper point and a dry cotton pellet. The orifice was closed with a cotton pellet and a temporary restorative material (MD Temp Plus, Meta Biomed, Cheongju, Korea). The canal filling state was confirmed by secondary micro-CT scanning. The proper filling state was determined by adherence of the paste to the canal walls with the absence of voids. If some specimens failed to have such filling quality, they were discarded and replaced by new specimens selected by the abovementioned methods. The specimens were stored at 37˚C in 100% relative humidity for one week.

## Irrigation procedures

After one week of intracanal medicament application, the temporary restorative material was removed, and calcium hydroxide was removed by one of the four protocols (Table 1) [8]. All canals were irrigated with 3% NaOCl solution at 2 mm shorter than the working length with up-and-down motions. A total volume of NaOCl solution used in irrigation by each protocol was 10 ml.

**Table 1. Irrigation systems used in the study.**

| Irrigation systems | | Tip characteristics | Tip sizes | Frequency | Irrigation methods [8] |
|---|---|---|---|---|---|
| Control | Needle irrigation | Side-vented needle (NaviTip, Ultradent Products, South Jordan, UT, USA) | 30G; straight | N/A | Canals were irrigated using a total of 10 mL of NaOCl for 60 seconds |
| EndoActivator (Dentsply Maillefer, Santa Barbara, CA, USA) | Sonic irrigation | Flexible, non-cutting polymer tip (Dentsply Maillefer, Santa Barbara, CA, USA) | #15/.02 | 166 Hz (10,000 cpm) | Canals were irrigated using a total of 10 mL of NaOCl, applying the device three times for 20 seconds each |
| EQ-S (Meta Systems, Seongnam, Korea) | Sonic irrigation | Flexible, non-cutting polymer tip (Meta Systems, Seongnam, Korea) | #15/.02 | 217 Hz (13,000 cpm) | Same as in EndoActivator |
| Vibringe (Cavex, Haarlem, Netherlands) | Sonic irrigation | Side-vented needle (NaviTip, Ultradent Products, South Jordan, UT, USA) | 30G; straight | 150 Hz | Same as in Control |

## Micro-CT analysis

The specimens were scanned using a micro-CT scanner at three time points: root curvature determination, after filling the canal with calcium hydroxide, and after removing the calcium hydroxide from the canal. The scanning parameters were the same for all three times of scans. The 3D images were acquired in the same manner at each time. The region of interest was the apical 3 mm of the root canal structure. Within the range of analysis, the volume of calcium hydroxide after filling and after removal was calculated. Percentage removal was determined according to the following equation:

$$Percentage\ removal\ of\ calcium\ hydroxide\ (\%)$$
$$= \frac{Volume\ of\ Ca(OH)_2\ after\ filling - volume\ of\ Ca(OH)_2\ after\ removal}{Volume\ of\ Ca(OH)_2\ after\ filling} \times 100$$

## Light-coupled tracking of oscillation

Each irrigation system was fixed so that it could stand alone and reveal its lateral face toward the camera. The tip of each irrigation system was 90 degrees to the camera at a 20 cm distance. The tips and the camera were set at the same height. The irrigation system was activated or left as is depending on the type. Ambient light was blocked with a cloth, and blue light from the LED unit (Elipar DeepCure-S, 3M ESPE, St. Paul, MN, USA) of 1,470 mW/cm$^2$ was shone onto the tip. As light was coupled along the tip to track its oscillating motion, real time images were collected. Also, the same images were obtained for tip oscillation within an artificial block with a curved (10°) canal. The entire procedure was repeated three times, each time with a new tip. Based on the motion traces of oscillating tips, maximum oscillating ranges were calculated with computer software (ImageJ, NIH, Bethesda, Maryland, MD, USA).

## Statistical analysis

For comparison of remaining volumes and removal percentages of calcium hydroxide among the irrigation systems and the root canal curvatures, the data did not follow a normal distribution, as confirmed by Shapiro-Wilk test. Thus, a nonparametric Kruskal-Wallis test and Mann-Whitney test with Bonferroni correction were used. All statistical analyses were performed at a significance level of 0.05 using IBM SPSS Statistics software Version 26.0 (IBM, Armonk, NY, USA).

## Results

The median percentage removal of intracanal medication from the part of apical 3 mm was shown in Table 2. In the straight canals, no significant differences existed among the groups,

**Table 2. Median removal percentages of intracanal medication.**

| Groups | Median Removal Percentage (%) | | |
|---|---|---|---|
| | Root canal curvatures | | |
| | **Straight** | **Moderate** | **Severe** |
| Group 1 Control | 99.95 [99.61, 100] [A] | 99.88 [94.55, 99.92] [AB] | 80.17 [71.05, 92.66] [A] |
| Group 2 Endoactivator | 100 [99.93, 100] [A] | 99.91 [99.51, 99.99] [A] | 99.53 [90.94, 99.84] [BC] |
| Group 3 EQ-S | 100 [97.77, 100] [A] | 99.95 [99.59, 100] [A] | 99.95 [99.38, 100] [C] |
| Group 4 Vibringe | 100 [99.19, 100] [A] | 96.19 [89.10, 99.32] [B] | 92.09 [86.21, 96.17] [AB] |

Values with the same subscripts are not significantly different compared within columns ($P > 0.05$)
Interquartile ranges [first quartile, third quartile] are shown in parentheses.

with all exhibiting nearly complete (100%) removal of calcium hydroxide. In the canals with severe curvature, the percentage removal was highest in Group 3 (EQ-S), followed by Group 2 (EndoActivator), Group 4 (Vibringe), and Group 1 (control). Significant differences existed between Group 1 and Groups 2 and 3, and between Group 3 and Group 4 ($p < 0.05$). The remaining volumes of calcium hydroxide in the canals of straight or moderate curvature were not significantly different among the groups (Table 3). Instead, significant differences existed in the canals based on severe curvature, in the opposite order as percentage removal ($p < 0.05$). Graphs in Fig 1 depicted the comparison among the different curvatures; all groups had significantly lower removal percentages and higher remaining volumes with severe curvature compared to non-severe curvature, except Group 3 ($p < 0.05$), which revealed no significant differences among the three curvatures.

Cross-sectional micro-CT images obtained at the apical 2-mm level (Fig 2) and 3D images (Fig 3) exhibited calcium hydroxide remaining in the canal space. In canals with severe curvature, remnants of calcium hydroxide were noticeable at the inner and outermost parts of the curvature in Groups 1 and 4.

The maximum oscillation range of each system is demonstrated in Fig 4. The lateral extent of oscillation was measured by light-coupled tracking of movement in free space and within the artificial canal block. Group 3 exhibited the largest extent of movement, followed by Group 2, while the needle tips in Groups 1 and 4 showed minimal movement (Figs 4 and 5A). Oscillation was restricted in the artificial block, resulting in near absence of movement for all groups (Fig 5B).

## Discussion

This study evaluated the efficacy of sonic irrigation systems for removing calcium hydroxide in the apical root canal. All three sonic systems exhibited removal capacity significantly higher

**Table 3. Median remaining volumes of intracanal medication.**

| Groups | Median Remaining Volume ($10^3$mm$^3$) | | |
|---|---|---|---|
| | Root canal curvatures | | |
| | **Straight** | **Moderate** | **Severe** |
| Group 1 Control | 0.83 [0, 1.84] [A] | 1.34 [0.40, 63.16] [A] | 94.60 [47.32, 156.86] [A] |
| Group 2 Endoactivator | 0 [0, 0.12] [A] | 0.42 [0.05, 1.91] [A] | 3.18 [1.27, 30.08] [BC] |
| Group 3 EQ-S | 0 [0, 9.99] [A] | 0.36 [0, 2.30] [A] | 0.72 [0, 1.95] [C] |
| Group 4 Vibringe | 0 [0, 2.44] [A] | 12.96 [3.18, 40.92] [A] | 26.38 [17.84, 81.99] [AB] |

Values with the same subscripts are not significantly different when compared within columns ($P > 0.05$)
Interquartile ranges [first quartile, third quartile] are shown in parentheses.

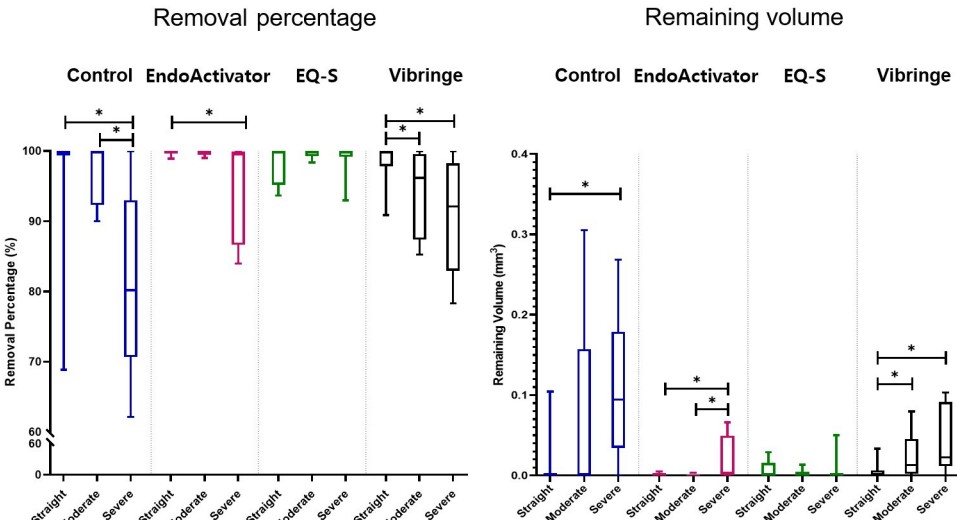

**Fig 1. Removal efficacy of intracanal medication in different curvatures and devices.** (a) Removal percentage of intracanal medication. (b) Remaining volume of intracanal medication. *: Values are significantly different within the same irrigation devices (p < 0.05).

than that of conventional needle irrigation. Also, free-oscillating ranges of the irrigation tips differed among the systems. The EQ-S showed the largest oscillating range and the highest percentage removal of calcium hydroxide. Therefore, the null hypothesis that removal of calcium hydroxide at the apical root canal would not be affected by the sonic irrigation system was rejected.

Previous studies evaluating canal irrigation methods have mainly focused on removal of debris and smear layer during chemomechanical cleaning of root canal systems. When root canal treatment is extended to multiple sessions, the canal space must be filled with interim

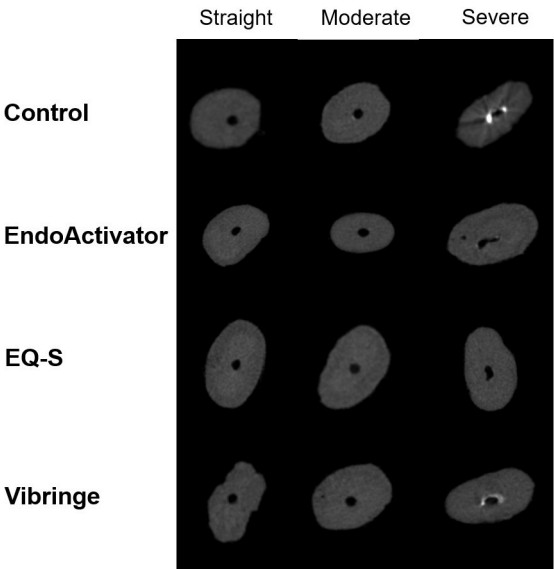

**Fig 2. Cross-sectional micro-CT images at 2 mm from the apical constriction.** The white mass inside the wall indicates remaining intracanal medication.

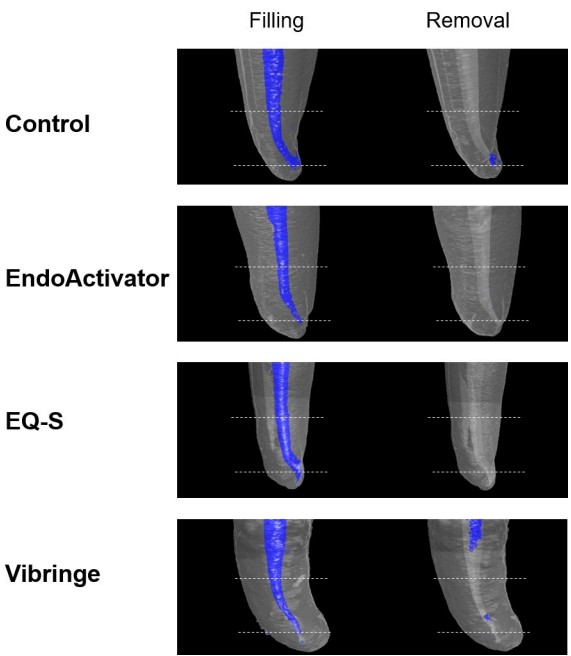

**Fig 3. 3D demonstration of intracanal medication within the severe curvatures after filling and after removal by different irrigation systems.** Intracanal medication is designated in blue, and the region of interest (3 mm from the apical constriction) is shown in dashed lines.

medicament to induce the pharmaceutical effect. At the time of final canal obturation, the medicament should be removed to expose dentinal tubules for application and penetration of sealers. Quantitative evaluation of cleaning efficacy often is presented as percentage removal of

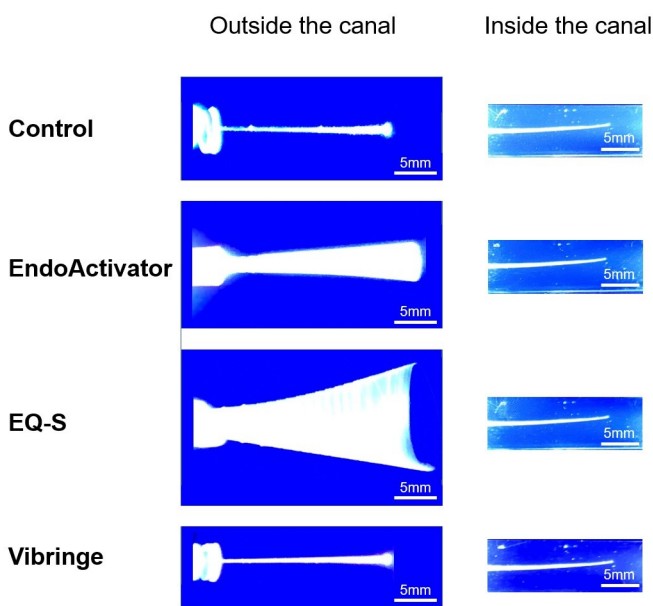

**Fig 4. Light-coupled tracking of the operating irrigation tips in free-range and inside the artificial block with a curved (10˚) canal.**

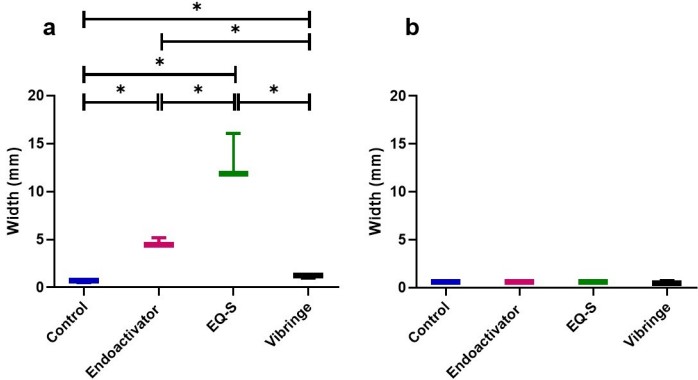

**Fig 5. Maximum oscillation widths of the light-coupled tips.** (a) Maximum oscillation widths outside the canal. (b) Maximum oscillation widths inside the artificial block with a curved (10˚) canal. *: Values are significantly different among the irrigation devices (p < 0.05).

intracanal medication as outcome values. Compared to the original volume of filled medication, the remaining volume was minimal, resulting in approximately 100% removal rate, as shown in the current study results. However, even traces of calcium hydroxide remnants adhering to the wall surface can negatively affect effective sealing of the intraradicular structure against microbial ingress from the oral cavity or the periapical tissues [24]. Another consideration is that the geometry of the canal is difficult to be standardized in the pooled samples of diverse configurations. Therefore, we performed an additional comparison of the remaining volumes of calcium hydroxide among the groups. The pre-packaged calcium hydroxide paste used in this study, Calcipex, is a water-based mixture with high flowability. A flexible and tapered plastic needle is attached to the syringe to allow the paste to easily flow into the space. In previous in vitro studies comparing the removal efficiency of Calcipex [25, 26], additional instrumentation or agitation was not applied after injecting of Calcipex paste. We confirmed the filling quality of the intracanal medication using micro-CT. The samples with proper adaptation of the paste to the canal walls and no detectable voids were selected for the further analysis. The serial images of micro-CT provided qualitative visualization and quantitative measurement of remaining calcium hydroxide. In cross-sectional images, apical canals with severe curvature resembled oval-shaped canals or an isthmus which is difficult to access by instruments and to flush out with irrigants. The three sonic systems revealed superior cleaning efficacy to conventional needle irrigation at the curved apex.

Many previous studies have evaluated the cleaning activity of various irrigation systems, often indiscriminately including PUI and sonic irrigation, and compared those with needle irrigation methods. In this study, 3% NaOCl solution was used as an irrigant during canal preparation and removal of intracanal medication. Calcium-chelating agent such as 17% EDTA solution may be used as a final rinse to remove the smear layer as recommended in clinical practice [27]. But it might not be a primary option as demonstrated in a systematic review evaluating ultrasonic irrigant activation and syringe irrigation [28]. In this study, we have focused on a broader scale of removal efficiency of calcium hydroxide using sonic irrigation systems, which is more practical in clinical circumstances. We compared three sonically activated systems with similar frequency (150–217 Hz), but different tip types: EndoActivator and EQ-S with flexible polymer tips and Vibringe with a stainless-steel needle tip. The aim of this study was to demonstrate the flexibility of each irrigation tip during oscillation using a light-tracking method. EQ-S had a more flexible and softer type of polymer tip with a larger extent

of movement compared to EndoActivator. When an irrigation needle or wire is incorporated into a narrow canal and vibrated, simultaneous contacts with rigid walls are inevitable. The irrigation tip would not be freely displaced, and fluid movements would be subjected to frictional forces between a boundary and vibrating medium to produce acoustic microstreaming [29]. The intensity of microstreaming is related directly to the streaming velocity and displacement that is proportional to the square of the displacement amplitude of the oscillating tips [5]. EQ-S had the largest range of oscillation, relating to more enhanced streaming effect for cleaning. The stainless-steel tip used with Vibringe yielded a minimal range of transversal movement, which might be related to its lower cleaning effect at the apical portion, particularly in severely curved canals. Our speculation was that limitedly flexible metal tips, as are often used in PUI, will be more highly affected by the confined geometry of canal walls. A previous study on file-to-wall contact during PUI showed that the file hit the wall and reversed production of a low-frequency secondary oscillation (6 kHz) that seemed to have more profound impact on activation of irrigant than did its primary oscillation (30 kHz) [7]. Another mechanism of ultrasonically or sonically activated irrigation was explained as acoustic cavitation that produces pressure fluctuations and nucleates vapor bubbles [5]. The intensity of cavitation is affected by several factors such as diameters and shapes of tips, oscillating amplitude, surrounding geometry, and properties of medium [29]. The softness and flexibility of irrigation tips seemed to be a critical part contributing to enhance vibration impact separately from the frequency and power of the oscillating instruments.

Another consideration in terms of clinical aspect is how the irrigation tip can be directed into the apical canal, particularly with narrow and curved configurations. In this study, we used single-rooted teeth, in which the tip would be more easily inserted than into multi-rooted teeth. Nevertheless, at the canal below the severe curvature, calcium hydroxide remnants were less completely washed out by a rigid stainless-steel tip, although the streaming was sonically-activated as with Vibringe. In clinical practice, insertion and retraction of irrigation tips deep into and out of a narrow and curved canal spaces are demanding procedural steps. Even when the tip can be readily inserted into the apical canal, clinicians might be concerned about the risk of tip separation similar to that when using the NiTi rotary files in curved narrow canals. In this light, flexible polymer tips of sonic irrigation systems have a major advantage over rigid metal tips equipped in other systems. And this could provide additional benefits for clinicians performing endodontic therapy in everyday practice.

Root canal geometry is variable based on individual anatomy of teeth. Clinical implications in endodontic research rely on representation of anatomical complexities and how relevantly specimens are assorted under experimental settings. In a systematic and critical review on measurement of root canal curvature, 3D imaging was valued over 2D microscopic evaluation to increase precision of both qualitative and quantitative measurements [30]. In evaluation of root curvature, we reviewed a stack of images from the sagittal view rotating 360° around the tooth axis and captured the most severe angulation of images. In terms of root curvature determination, this study was based on a methodology with high accuracy. Still, to obtain highly standardized specimens, root canal models must have uniform severity of curvatures and simulate anatomical complexities such as anastomoses and ramifications. Also, using a light-transmissible medium for tooth models will incorporate more easily a sonoluminescence technique to capture the oscillating modalities. These experimental setups are obtainable by current techniques of computer-aided designing and manufacturing. Enhancement of debridement and disinfection without damage to anatomical structure is a goal of activated irrigation systems. With standardized root canal models, the integrity of radicular structures will be confirmed accurately before and after irrigation procedures. This provides another topic of interest for future studies.

## Conclusions

The three sonically activated irrigation systems used in this study showed increased removal capacity of calcium hydroxide in the apical root canal compared to conventional needle irrigation. EQ-S had an extended range of oscillation with a flexible irrigation tip and higher cleaning capacity at the curved apex compared to other sonic irrigation systems. Sonically activated irrigation systems with a flexible tip can be beneficial for removal of intracanal dressing materials in the curved apical canals.

## Supporting information

**S1 Data.**
(XLSX)

## Author Contributions

**Conceptualization:** Shin Hye Chung.

**Data curation:** Wonjoon Moon.

**Formal analysis:** Wonjoon Moon.

**Funding acquisition:** Shin Hye Chung.

**Investigation:** Wonjoon Moon, Juhea Chang.

**Project administration:** Shin Hye Chung, Juhea Chang.

**Validation:** Shin Hye Chung, Juhea Chang.

**Visualization:** Wonjoon Moon.

**Writing – original draft:** Wonjoon Moon, Juhea Chang.

**Writing – review & editing:** Shin Hye Chung, Juhea Chang.

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
