## [Decision Letter · Decision Letter 0]

16 Mar 2022

PONE-D-21-30735

Sonic irrigation for removal of calcium hydroxide in the apical root canal: A micro-CT and light-coupled tracking analysis

PLOS ONE

Dear Dr. Juhea Chang,

Thank you for submitting your manuscript to PLOS ONE. After careful consideration, we feel that it has merit but does not fully meet PLOS ONE’s publication criteria as it currently stands. Therefore, we invite you to submit a revised version of the manuscript that addresses the points raised during the review process.

Please revise your manuscript following the reviewers' comments.

We look forward to receiving your revised manuscript.

Kind regards,

Zhaoqiang Zhang

Academic Editor

PLOS ONE

2. Please provide additional details regarding participant consent. In the Methods section,

please ensure that you have specified (1) whether consent was informed and (2) what type you obtained

(for instance, written or verbal). If your study included minors, state whether you obtained consent from parents

 or guardians. If the need for consent was waived by the ethics committee, please include this information.

3. PLOS requires an ORCID iD for the corresponding author in Editorial Manager on papers submitted after December 6th, 2016. Please ensure that you have an ORCID iD and that it is validated in Editorial Manager. To do this, go to ‘Update my Information’ (in the upper left-hand corner of the main menu), and click on the Fetch/Validate link next to the ORCID field. This will take you to the ORCID site and allow you to create a new iD or authenticate a pre-existing iD in Editorial Manager. Please see the following video for instructions on linking an ORCID iD to your Editorial Manager account: https://www.youtube.com/watch?v=_xcclfuvtxQ.

Reviewers' comments:

Reviewer's Responses to Questions

**Comments to the Author**

1. Is the manuscript technically sound, and do the data support the conclusions?

Reviewer #1: Yes

Reviewer #2: Yes

2. Has the statistical analysis been performed appropriately and rigorously? 

Reviewer #1: Yes

Reviewer #2: Yes

3. Have the authors made all data underlying the findings in their manuscript fully available?

Reviewer #1: Yes

Reviewer #2: Yes

4. Is the manuscript presented in an intelligible fashion and written in standard English?

Reviewer #1: No

Reviewer #2: No

5. Review Comments to the Author

Reviewer #1: This manuscript constitutes an attempt to evaluate the efficacy of three different sonic irrigation agitation systems on calcium hydroxide removal from apical third of the root canals using micro-CT. The study was well performed and clinically relevant. However, English needs to be improved. Some of the specific queries have been addressed below.

*Introduction:

- Line 49- the sentence- (Remnants of calcium hydroxide…filling materials)- needs reference.

- In the aim of the study, mention the three different types of sonic irrigation systems used.

*Methodology:

- How was sample size calculated?

- Posterior teeth with anastomosis would have been more relevant than using single rooted teeth.

- How were the teeth cleaned and stored?

- How was working length determined?

- In the beginning, it is mentioned 96 teeth were used. However, later it is mentioned 36 teeth were subjected to 3 categories based on root canal curvature. Its not clear, exactly what was the sample size used. Mention in detail.

- Why canals were not irrigated during instrumentation?

- How was smear layer removed which was formed during instrumentation?

- Mention the manufacturer’s details of paper points and NaOCl.

- What was the volume of calcium hydroxide placed inside each sample?

- How was calcium hydroxide placed inside each root canal?

- In control and vibringe groups, what was the time period of irrigation agitation?

- A chelator should have been used along with NaOCl to remove calcium hydroxide efficiently.

- Mention the details of calcium hydroxide paste used.

- How was normality of data evaluated?

- Do not insert tables and figure legends in between the text.

*Results:

- Results of the volume of calcium hydroxide in all the samples in different groups before removal has to be mentioned to ensure uniform distribution of calcium hydroxide.

Reviewer #2: General Comments: Thank you for your submission to the Plos One. The purpose of the present study was to evaluate the efficacy of three sonic irrigation systems for removal of calcium hydroxide dressing from the apical root canal. The premise of the study is sound, and the article is well written. Furthermore, the experimental procedures are properly delineated; and it has merits to be published in the Plos One. However, there are some drawbacks which prevent its publication in the current form. The article needs a major revision by authors prior acceptance for publication.

- “Sonically activated irrigation systems with a flexible tip can be beneficial for cleaning of intracanal medication in the curved apical canals.” Please, rephrase it as suggested: “Sonically activated irrigation systems with a flexible tip can be beneficial for calcium hydroxide intracanal dressing removal in curved apical root canals.”

- “Calcium hydroxide is the most commonly used as intracanal medicament during endodontic treatment because of its physical and biological advantages such as antibacterial effect, tissue dissolving, promoting hard tissue formation, reducing bacterial toxic products, and healing periapical tissues (1).” This sentence is too long and confusing. Please, rephrase it.

- “This limited cleaning effect is more pronounced below the root canal curvature narrowing to the apical constriction. Considering the three-dimensional complexities of the apical third of the root canal system, clinical outcomes can be evaluated based on both qualitative and quantitative measurements.” References are missing for these two sentences.

- The originality, the state of the art of the studied subject and the scientific contribution of the study were not clear.

- Kindly check the grammar and written style of the whole manuscript.

- The sample size calculation was not described.

- According to the authors, “The curvatures of each root curvatures were determined in sagittal views from the micro-CT images.” However, as stated at the next sentence, “The curvatures were calculated with the Schneider method (17).” The authors should explain such a procedure with more details for better understanding by the readers.

- Did the authors perform a micro-CT analysis considering the anatomical features of the teeth for specimens grouping?

- The randomization of the specimen’s distribution into the experimental groups should be clearly described in the main text.

- The manufacturer details of the irrigation systems are missing.

- What was the working length of the root canals? Please, add this information.

- “…..and filled with calcium hydroxide paste (Calcipex II, Nippon Shika Yakuhin, Shimonoseki, Japan), a dry cotton pellet, and a temporary restorative material (MD Temp Plus, Meta Biomed, Cheongju, Korea).” This sentence does not make sense.

- “The canal filling state was confirmed by the secondary micro-CT scanning.” What variables were considered to confirm the filling quality? Were specimens discarded from the final sample after such analysis?

- The correct form is 3% NaOCl “solution”!!! Please, check the whole text.

- The way in which the examiners were trained/calibrated and the result of inter- and intra-examiner agreement are not shown.

- The statistical analysis should be described with more details. How were sample normality and homogeneity tested?

- According to the authors “However, even traces of calcium hydroxide remnants adhering to the wall surface can negatively affect hermetic sealing of the intraradicular

structure against microbial ingress from the oral cavity or the periapical tissues (18).” Hermetic sealing is a falacy. Please, change this term.

- The Conclusion Section should be more concise, and for this reason, it must be rewritten for better understanding by the readers.

6. PLOS authors have the option to publish the peer review history of their article (what does this mean?). If published, this will include your full peer review and any attached files.

Reviewer #1: No

Reviewer #2: No

---

## [Author Response · Author response to Decision Letter 0]

4 May 2022

Authors ‘responses to the reviewers

Title: Sonic irrigation for removal of calcium hydroxide in the apical root canal: A micro-CT and light-coupled tracking analysis

Reviewer #1: 

This manuscript constitutes an attempt to evaluate the efficacy of three different sonic irrigation agitation systems on calcium hydroxide removal from apical third of the root canals using micro-CT. The study was well performed and clinically relevant. However, English needs to be improved. Some of the specific queries have been addressed below.

*Introduction:

#1-1. Line 49- the sentence- (Remnants of calcium hydroxide…filling materials)- needs reference.

Author response to #1-1

We newly added the following reference (#4) at the end of the above-mentioned sentence.

Ref #4 Kim SK, Kim YO. Influence of calcium hydroxide intracanal medication on apical seal. Int Endod J. 2002 Jul;35(7):623-8. 

#1-2. In the aim of the study, mention the three different types of sonic irrigation systems used.

Author response to #1-2

We rewrote the aim of the sentence as follows:

“In this study, we applied conventional needle irrigation and three types of sonic irrigation systems (EndoActivator, EQ-S, and Vibringe) for removal of intracanal calcium hydroxide and evaluated the volumes of calcium hydroxide remaining in the apical space among different root canal curvatures using micro-CT.”

*Methodology:

#1-3. How was sample size calculated?

Author response to #1-3

We referred to a previous study on calcium hydroxide removal using PUI that used n = 8 per each group. The following sentence was inserted in the Materials and Methods.

“The sample size was determined based on a previous study that reported means and SDs for removal efficacy of calcium hydroxide using micro-CT imaging (Ref #23).”

Ref #23. Silva LJM, Pessoa OF, Teixeira MBG, Gouveia CH, Braga RR. Micro-CT evaluation of calcium hydroxide removal through passive ultrasonic irrigation associated with or without an additional instrument. Int Endod J. 2014;48(8):768-73.

#1-4. Posterior teeth with anastomosis would have been more relevant than using single rooted teeth.

Author response to #1-4

We agree with the reviewer’s point that the isthmus between two canals is difficult to access and would require more competency of intracanal irrigation. In this study, we sorted natural tooth specimens into three different types of root canal curvatures and performed volumetric evaluations of the canal spaces before and after cleaning and removal of intracanal medication. For comparison of removal efficacy, anatomical irregularities had to be minimized among the included specimens. Therefore, the anatomical uniqueness and complexity had to be avoided as much as possible for our quantitative analysis. Agreeing with the reviewer’s comments, we rewrote the related sentence in the Discussion.

“Still, to obtain highly standardized specimens, we will consider constructing root canal models with a uniform severity of curvature.” was rewritten as follows:

“Still, to obtain highly standardized specimens, root canal models must have uniform severity of curvatures and simulate anatomical complexities such as anastomoses and ramifications.”

#1-5. How were the teeth cleaned and stored?

Author response to #1-5

We rewrote the related sentence as follows:

“The roots of the teeth were ultrasonically cleaned and stored in 0.9 g/L thymol solution (Sigma-Aldrich, St. Louis, MO, USA) (Ref #21).” 

Ref #21. Yang Q, Liu MW, Zhu LX, Peng B. Micro‐CT study on the removal of accumulated hard‐tissue debris from the root canal system of mandibular molars when using a novel laser‐activated irrigation approach. Int Endod J. 2020;53(4):529-38.

#1-6. How was working length determined?

Author response to #1-6

We added more description about working length determination and the relevant reference as follows: 

“Canal patency was confirmed with a #K-10 file (K-file, Maillefer Instruments, Ballaigues, Switzerland) until the tip was just visible at the apical foramen (Ref # 18). The working lengths were determined as 1 mm less than that length.”

Ref #18. Yang Q, Liu MW, Zhu LX, Peng B. Micro‐CT study on the removal of accumulated hard‐tissue debris from the root canal system of mandibular molars when using a novel laser‐activated irrigation approach. Int Endod J. 2020;53(4):529-38.

#1-7. In the beginning, it is mentioned 96 teeth were used. However, later it is mentioned 36 teeth were subjected to 3 categories based on root canal curvature. Its not clear, exactly what was the sample size used. Mention in detail.

Author response to #1-7

Thanks for the pointing out the error. We amended the sentence as follows.

“Considering their root curvatures, the 96 teeth were assigned into one of three categories: straight (0-5°), moderate (6-20°), and severe (> 21°) (n = 32/category). The 32 teeth in each category were then randomly distributed into four groups of irrigation systems: Group 1 (control), Group 2 (EndoActivator), Group 3 (EQ-S), and Group 4 (Vibringe) (n = 8/group).”

#1-8. Why canals were not irrigated during instrumentation?

Author response to #1-8

We appreciate for the reviewer’s comment pointing out the incomplete descriptions of the experimental procedures. The following sentence was newly inserted: 

“During instrumentation, the root canals were irrigated with 2 mL 1% NaOCl solution at each change of instrumentation.”

The Materials and Methods section was substantiated with additional statements describing the procedures in detail. Please see the response to #1-9. 

#1-9. How was smear layer removed which was formed during instrumentation?

Author response to #1-9

A chelating agent such as 15-17% EDTA is generally recommended as a final rinse to remove the smear layer prior to canal obturation. In addition, many in vitro studies evaluating the effectiveness of irrigating methods tended to use EDTA as a conjunctive irrigant, particularly for microscopic confirmation of the microbial and physical cleanliness of the dentinal tubules. In a systematic review of 48 in vitro studies comparing ultrasonic irrigant activation and syringe irrigation (Ref #28), NaOCl (1-10%) was the most frequently used irrigatns (46 studies), while EDTA was used in fewer studies (21 studies). In this study, we have focused on the broad-range of removal efficiency of calcium hydroxide using sonic irrigation systems, which is a practical issue in clinical circumstances. Still, we understand the reviewer’s concern, and added the following sentence in the Discussion.

“In this study, 3% NaOCl solution was used as an irrigant during canal preparation and removal of intracanal medication. Calcium-chelating agent such as 17% EDTA solution may be used as a final rinse to remove the smear layer as recommended in clinical practice (Ref #27). However, it might not be a primary option, as demonstrated in a systematic review evaluating ultrasonic irrigant activation and syringe irrigation (Ref #28). In this study, we have focused on a broader scale of removal efficiency of calcium hydroxide using sonic irrigation systems, which is a practical issue in clinical circumstances.”

Ref #27. Haapasalo M, Shen Y, Wang Z, Gao Y. Irrigation in endodontics. Br Dent J. 2014 Mar;216(6):299-303. 

Ref #28. Căpută PE, Retsas A, Kuijk L, Chávez de Paz LE, Boutsioukis C. Ultrasonic Irrigant Activation during Root Canal Treatment: A Systematic Review. J Endod. 2019 Jan;45(1):31-44.e13. 

#1-10. Mention the manufacturer’s details of paper points and NaOCl.

Author response to #1-10

The manufacturer’s details were inserted as follows:

“After canal preparation, the specimens were dried with paper points (Absorbent Paper Points, Meta Biomed, Cheongju, Korea).”

#1-11. What was the volume of calcium hydroxide placed inside each sample?

Author response to #1-11

We substantiated the required information as follows.

“The 0.1 mL of calcium hydroxide paste (Calcipex II, Nippon Shika Yakuhin, Shimonoseki, Japan) was injected slowly with minimal pressure to fill the space.”

#1-12. How was calcium hydroxide placed inside each root canal?

Author response to #1-12

We added the following sentences and references in the Discussion. 

“The pre-packaged calcium hydroxide paste used in this study, Calcipex, is a water-based mixture with high flowability. A flexible and tapered plastic needle is attached to the syringe to allow the paste to easily flow into the space. In previous in vitro studies comparing the removal efficiency of Calcipex (Ref #25, #26), additional instrumentation or agitation was not applied after injecting of the Calcipex paste.”

Ref #25. Kim T, Kim MA, Hwang YC, Rosa V, Del Fabbro M, Min KS. Effect of a calcium hydroxide-based intracanal medicament containing N-2-methyl pyrrolidone as a vehicle against Enterococcus faecalis biofilm. J Appl Oral Sci. 2020 Mar 27;28:e20190516. 

Ref #26. Lim MJ, Jang HJ, Yu MK, Lee KW, Min KS. Removal efficacy and cytotoxicity of a calcium hydroxide paste using N-2-methyl-pyrrolidone as a vehicle. Restor Dent Endod. 2017 Nov;42(4):290-300. 

In addition, the following sentence was inserted in the Materials and Methods and in the Discussion.

“The proper filling state was determined by the adherence of the paste to the canal walls and absence of voids.”

“The samples with proper adaptation of the paste to the canal walls and no detectable voids were selected for the further analysis.”

#1-13. In control and vibringe groups, what was the time period of irrigation agitation?

Author response to #1-13

As in the other groups, the Control and Vibringe groups were irrigated for 60 seconds. We revised Table 1 accordingly and mentioned the reference (Ref #8).

#1-14. A chelator should have been used along with NaOCl to remove calcium hydroxide efficiently.

Author response to #1-14

We understand the reviewer’s concern. Please refer to the response to #1-9

#1-15. Mention the details of calcium hydroxide paste used.

Author response to #1-15

We added the following sentence with a new reference in the Materials and Methods.

“The calcium hydroxide paste used in this study consisted of calcium hydroxide, barium sulfate, and distilled water and is easy to handle as a root canal filling material (Ref #24).”

Ref #24. Hosoya N, Kurayama H, Iino F, Arai T. Effects of calcium hydroxide on physical and sealing properties of canal sealers. Int Endod J. 2004;37(3):178-84.

#1-16. How was normality of data evaluated?

Author response to #1-16

We added a description about the normal distribution of the data as follows:

“For comparison of remaining volumes and removal percentages of calcium hydroxide among the irrigation systems and root canal curvatures, the data did not follow a normal distribution, as confirmed by the Shapiro-Wilk test.”

#1-17. Do not insert tables and figure legends in between the text.

Author response to #1-17

We removed the inserted tables and figures from the text. However, in the submission guideline shown below, we were asked to insert figure captions in the text. Based on the editorial office requirement for authors, we will be happy to revise it accordingly.

“Authors’ guideline: Figure captions must be inserted in the text of the manuscript, immediately following the paragraph in which the figure is first cited (read order).”

*Results:

#1-18. Results of the volume of calcium hydroxide in all the samples in different groups before removal has to be mentioned to ensure uniform distribution of calcium hydroxide.

Author response to #1-18

We additionally mentioned the volume of calcium hydroxide injected. Please refer to the response to #1-11.

Reviewer #2:

 General Comments: Thank you for your submission to the Plos One. The purpose of the present study was to evaluate the efficacy of three sonic irrigation systems for removal of calcium hydroxide dressing from the apical root canal. The premise of the study is sound, and the article is well written. Furthermore, the experimental procedures are properly delineated; and it has merits to be published in the Plos One. However, there are some drawbacks which prevent its publication in the current form. The article needs a major revision by authors prior acceptance for publication.

#2-1. “Sonically activated irrigation systems with a flexible tip can be beneficial for cleaning of intracanal medication in the curved apical canals.” Please, rephrase it as suggested: “Sonically activated irrigation systems with a flexible tip can be beneficial for calcium hydroxide intracanal dressing removal in curved apical root canals.”

Author response to #2-1

We appreciate the reviewer’s input. The sentence was corrected as suggested.

#2-2. “Calcium hydroxide is the most commonly used as intracanal medicament during endodontic treatment because of its physical and biological advantages such as antibacterial effect, tissue dissolving, promoting hard tissue formation, reducing bacterial toxic products, and healing periapical tissues (1).” This sentence is too long and confusing. Please, rephrase it.

Author response to #2-2

We amended the sentence as follows:

“Calcium hydroxide is the most common intracanal medicament used during endodontic treatment (1). It has diverse physical and biological advantages such as antibacterial effects, ability to dissolve tissue, promotion of hard tissue formation, reduction of bacterial toxic products, and healing of periapical tissues (2).”

#2-3. “This limited cleaning effect is more pronounced below the root canal curvature narrowing to the apical constriction. Considering the three-dimensional complexities of the apical third of the root canal system, clinical outcomes can be evaluated based on both qualitative and quantitative measurements.” References are missing for these two sentences.

Author response to #2-3

We cited the following references related to the above-mentioned sentences:

Ref #12. Caron G, Nham K, Bronnec F, Machtou P. Effectiveness of different final irrigant activation protocols on smear layer removal in curved canals. J Endod. 2010 Aug;36(8):1361-6. 

Ref #13. Kirmizi D, Aksoy U, Orhan K. Efficacy of Laser-Activated Irrigation and Conventional Techniques in Calcium Hydroxide Removal from Simulated Internal Resorption Cavities: Micro-CT Study. Photobiomodul Photomed Laser Surg. 2021 Oct;39(10):674-681. 

#2-4. The originality, the state of the art of the studied subject and the scientific contribution of the study were not clear.

Author response to #2-4

We agree with the reviewer’s point that the research interest was unclear. There are many previous studies regarding activation systems for root canal irrigation. Ultrasonic, passive ultrasonic, and sonic irrigation systems were indiscriminately compared among the different systems. We attempted to compare the devices with a similar range of working frequency, focusing on the characteristics of the irrigation tips. This was why we measured the extent of the movement of the irrigation tips. We clarify our research interest and background using the following sentences in the Discussion. 

“Many previous studies have evaluated the cleaning activity of various irrigation systems, often indiscriminately including PUI and sonic irrigation, and compared those with needle irrigation methods.”

“The aim of this study was to demonstrate the flexibility of each irrigation tip during oscillation using a light-tracking method.”

 “In clinical practice, insertion and retraction of irrigation tips deep into and out of narrow and curved canal spaces are demanding procedural steps…. And this could provide additional benefits for clinicians performing endodontic therapy in everyday practice.”

#2-5. Kindly check the grammar and written style of the whole manuscript.

Author response to #2-5

We received English proof reading and editing of the revised manuscript through a language editing service and corrected accordingly. The editorial certificate was uploaded with the revised documents.

#2-6. The sample size calculation was not described.

Author response to #2-6

Please refer to response #1-3.

#2-7. According to the authors, “The curvatures of each root curvatures were determined in sagittal views from the micro-CT images.” However, as stated at the next sentence, “The curvatures were calculated with the Schneider method (17).” The authors should explain such a procedure with more details for better understanding by the readers.

Author response to #2-7

We modified the following sentences for clarification.

“Micro-CT scans (Skyscan 1172, Bruker, Kontich, Belgium) were performed to obtain sagittal images of each root. The scanning parameters were 100kV and 100μA at the Al + Cu filter with an exposure time of 632ms. The pixel size was 30 μm with a rotation of 0.70 and an average frame number of three. The 3D images were acquired by 3D reconstruction (NRecon, Bruker, Kontich, Belgium), modeling (CTAn, Bruker, Kontich, Belgium), and analysis (CTVol, Bruker, Kontich, Belgium). Based on the sagittal images of each root, the curvatures were calculated with the Schneider method (22).”

#2-8. Did the authors perform a micro-CT analysis considering the anatomical features of the teeth for specimens grouping?

Author response to #2-8

Yes. The first micro-CT analysis was performed to obtain refined sagittal images of the roots to calculate accurate root curvature. Please refer to the author response to #2-7.

#2-9. The randomization of the specimen’s distribution into the experimental groups should be clearly described in the main text.

Author response to #2-9

We clarified the random distribution process:

“Considering their root curvatures, the 96 teeth were assigned into one of three categories: straight (0-5°), moderate (6-20°), and severe (> 21°) (n = 32/category). The 32 teeth in each category were then randomly distributed into four groups of irrigation systems: Group 1 (control), Group 2 (EndoActivator), Group 3 (EQ-S), and Group 4 (Vibringe) (n = 8/group).”

#2-10. The manufacturer details of the irrigation systems are missing.

Author response to #2-10

As suggested, we provided the manufacturer’s details in Table 1. 

#2-11. What was the working length of the root canals? Please, add this information.

Author response to #2-11

“Canal patency was confirmed with a #K-10 file (K-file, Maillefer Instruments, Ballaigues, Switzerland) until the tip was just visible at the apical foramen (Ref #18). The working lengths were determined as 1 mm less than that length.”

Ref #18. Yang Q, Liu MW, Zhu LX, Peng B. Micro‐CT study on the removal of accumulated hard‐tissue debris from the root canal system of mandibular molars when using a novel laser‐activated irrigation approach. Int Endod J. 2020;53(4):529-38.

#2-12. “…..and filled with calcium hydroxide paste (Calcipex II, Nippon Shika Yakuhin, Shimonoseki, Japan), a dry cotton pellet, and a temporary restorative material (MD Temp Plus, Meta Biomed, Cheongju, Korea).” This sentence does not make sense.

Author response to #2-12

We substantiated the materials and methods used in canal filling with the following sentence:

“After canal preparation, the canal space was dried with paper points (Absorbent Paper Points, Meta Biomed, Cheongju, Korea). The tip of the syringe containing paste was inserted into the canal to 1 mm shorter than the binding point (Ref #23). The 0.1 mL of calcium hydroxide paste (Calcipex II, Nippon Shika Yakuhin, Shimonoseki, Japan) was injected slowly with minimal pressure to fill the space… The medicament was then condensed using a paper point and a dry cotton pellet. The orifice was closed with a cotton pellet and a temporary restorative material (MD Temp Plus, Meta Biomed, Cheongju, Korea).”

Ref #23. Peters CI, Koka RS, Highsmith S, Peters OA. Calcium hydroxide dressings using different preparation and application modes: density and dissolution by simulated tissue pressure. Int Endod J. 2005 Dec;38(12):889-95. 

#2-13. “The canal filling state was confirmed by the secondary micro-CT scanning.” What variables were considered to confirm the filling quality? Were specimens discarded from the final sample after such analysis?

Author response to #2-13

We added the following sentences to better explain the process:

“The proper filling state was determined by adherence of the paste to the canal walls with the absence of voids. If some specimens failed to have such filling quality, they were discarded and replaced by new specimens selected by the abovementioned methods.”

#2-14. The correct form is 3% NaOCl “solution”!!! Please, check the whole text.

Author response to #2-14

We corrected it throughout the entire manuscript.

#2-15. The way in which the examiners were trained/calibrated and the result of inter- and intra-examiner agreement are not shown.

Author response to #2-15

We agree that the examiner training/calibration and inter-/intra-examiner agreements are essential information for evaluations based on ratings. However, as our analysis was based on quantitative calculations performed by the computer software, we thought that we could omit such information. All the experiments and calculations were performed by a single experienced operator. 

#2-16 The statistical analysis should be described with more details. How were sample normality and homogeneity tested?

Author response to #2-16

Please refer to the response #1-16.

#2-17. According to the authors “However, even traces of calcium hydroxide remnants adhering to the wall surface can negatively affect hermetic sealing of the intraradicular

structure against microbial ingress from the oral cavity or the periapical tissues (18).” Hermetic sealing is a falacy. Please, change this term.

Author response to #2-17

We agree with the reviewer’s comment. The above-mentioned term was corrected as follows:

“… can negatively influence effective sealing of….”

#2-18. The Conclusion Section should be more concise, and for this reason, it must be rewritten for better understanding by the readers.

Author response to #2-18

The paragraph in the Conclusions was re-written for more clarification as suggested.

“Sonically activated irrigation systems exhibited increased removal capacity of calcium hydroxide in the apical root canal compared to conventional needle irrigation. EQ-S had an extended range of oscillation with a flexible irrigation tip and higher cleaning capacity at the curved apex compared to other sonic irrigation systems.” 

“The three sonically activated irrigation systems used in this study showed increased removal capacity of calcium hydroxide in the apical root canal compared to conventional needle irrigation. EQ-S had an extended range of oscillation with a flexible irrigation tip and higher cleaning capacity at the curved apex compared to other sonic irrigation systems. Sonically activated irrigation systems with a flexible tip can be beneficial for removal of intracanal dressing materials in the curved apical canals.”

Author response:

Using the PACE system, we adjusted all the figures.

---

## [Decision Letter · Decision Letter 1]

9 May 2022

Sonic irrigation for removal of calcium hydroxide in the apical root canal: A micro-CT and light-coupled tracking analysis

PONE-D-21-30735R1

Dear Dr. Juhea Chang,

We’re pleased to inform you that your manuscript has been judged scientifically suitable for publication and will be formally accepted for publication once it meets all outstanding technical requirements.

Kind regards,

Zhaoqiang Zhang

Academic Editor

PLOS ONE

Additional Editor Comments (optional):

Reviewers' comments:

Reviewer's Responses to Questions

**Comments to the Author**

1. If the authors have adequately addressed your comments raised in a previous round of review and you feel that this manuscript is now acceptable for publication, you may indicate that here to bypass the “Comments to the Author” section, enter your conflict of interest statement in the “Confidential to Editor” section, and submit your "Accept" recommendation.

Reviewer #1: All comments have been addressed

Reviewer #2: All comments have been addressed

2. Is the manuscript technically sound, and do the data support the conclusions?

Reviewer #1: Yes

Reviewer #2: Yes

3. Has the statistical analysis been performed appropriately and rigorously? 

Reviewer #1: Yes

Reviewer #2: Yes

4. Have the authors made all data underlying the findings in their manuscript fully available?

Reviewer #1: Yes

Reviewer #2: Yes

5. Is the manuscript presented in an intelligible fashion and written in standard English?

Reviewer #1: Yes

Reviewer #2: Yes

6. Review Comments to the Author

Reviewer #1: Dear Authors,

Thank you for revising your manuscript appropriately based on the referees comments.

Thank Youi

Reviewer #2: The authors performed all the suggested modifications. The improvement in the manuscript is remarkable. Therefore, I would like to recommend the publication of this manuscript.

7. PLOS authors have the option to publish the peer review history of their article (what does this mean?). If published, this will include your full peer review and any attached files.

Reviewer #1: No

Reviewer #2: No

---

## [Editor Report · Acceptance letter]

23 May 2022

PONE-D-21-30735R1 

Sonic irrigation for removal of calcium hydroxide in the apical root canal: A micro-CT and light-coupled tracking analysis 

Dear Dr. Chang:

I'm pleased to inform you that your manuscript has been deemed suitable for publication in PLOS ONE. Congratulations! Your manuscript is now with our production department. 

Kind regards, 

on behalf of

Dr. Zhaoqiang Zhang 

Academic Editor

PLOS ONE